# Mirages of Misalignment: How Superposition Distorts Neural Representation Geometry

## Abstract

Neural networks trained on the same tasks achieve similar performance, but this is not always reflected in their measured representational alignment. We propose that this discrepancy arises from superposition or mixed selectivity, where individual neurons represent mixtures of features. Consequently, two networks representing an identical set of features can appear dissimilar if their neurons mix those features differently. This may explain why higher-dimensional networks, which are less prone to compressing mixtures of features, often show better alignment than smaller models with greater behavioral similarity. We formalize this through an analytic theory predicting apparent misalignment for common linear metrics like Representational Similarity Analysis (RSA) and Linear Regression, validating it from random projections to real neural networks. Using sparse autoencoders and K-Means to extract disentangled features while controlling for dimensionality, we find that feature-based alignment reveals higher similarity, particularly for early and lower-dimensional regions. Some comparisons show decreased alignment with disentanglement, and RSA and Linear Regression often disagree in these cases. Simulations predict that higher RSA relative to Linear Regression in neural space indicates shared inductive biases—a pattern confirmed in real data. Our results demonstrate that superposition and dimensionality interactions obscure the true alignment of lower-dimensional systems, while feature-based alignment allows us to more directly interrogate performance-relevant sources of misalignment, with important implications for model selection.

## 1 Introduction

The development of deep neural networks capable of human-level performance on tasks such as object recognition and natural language has prompted a fundamental question: do different neural systems converge to similar representations (Rumelhart et al., 1986; Goldstein et al., 2022; Peterson et al., 2018; Sucholutsky et al., 2023; Huh et al., 2024; Reizinger et al., 2024)? Answering this requires comparing representations across models with varied architectures, training data, and objectives, a challenge central to ideas like the platonic representation hypothesis (Huh et al., 2024; Reizinger et al., 2025). To measure these similarities, researchers turn to alignment metrics such as Representational Similarity Analysis (RSA) (Kriegeskorte & Wei, 2021) which abstract away from individual neurons to compare the geometry of population-level activity. Alternatively, Linear Regression is also used which learns a linear map to predict one network's activity from another. Both metrics have become powerful alignment tools, yielding remarkable insights into shared structure (Yamins et al., 2014; Khaligh-Razavi & Kriegeskorte, 2014; Cadena et al., 2019; Khosla et al., 2021; Schrimpf et al., 2021; Conwell et al., 2024; Prince et al., 2024). However, the neural networks with highest alignment scores are not always the most behaviorally (e.g., task performance) or mechanistically (e.g., sharing computational strategies) similar, leading to low performance-alignment correspondence (Schaeffer et al., 2024). This prompts the question: do behaviorally-similar models truly arrive at distinct representational solutions, or do confounding factors obscure the true representational similarities captured by standard metrics?

We propose the performance-alignment gap arises from *superposition* (or *mixed selectivity*), where individual neurons represent mixtures of multiple independent features (Smolensky, 1990; Elhage et al., 2022; Klindt et al., 2025). In this regime, neural networks can *linearly* represent more features than they have neurons by distributing features across overlapping neural codes. Consequently,

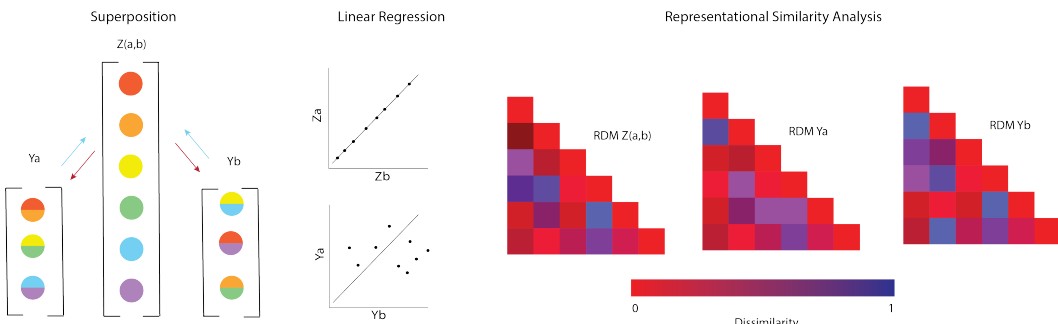

Figure 1: **Illustration of Core Idea. Superposition:** Two neural networks share an identical set of latent features ($Z_a = Z_b$), but compress them (red arrows) in different ways $Y_a \neq Y_b$. Thus, computing alignment over the raw neural activations of network A ($Y_a$) and B ($Y_b$) leads to low representational similarity of these networks. We propose using sparse dictionary learning to recover (blue arrows) the shared features of networks from their raw activations prior to using alignment metrics (Donoho, 2006). **Linear regression:** Assuming perfect latent recovery, the maximum pairwise correlation between latent activations is $1.0$, and will be greater than the correlation between raw neural activations. **Representational similarity analysis:** Rather than directly correlating neural (or latent) activation, RSA first computes pairwise (dis)similarity matrices of neural responses to features. Depicted are representational similarity matrices (or their dissimilarity counterparts), which are correlated to produce an alignment score. As with linear regression, the RSA score for perfectly recovered latents is $1.0$, and greater than the RSA score over neural activations.

two networks could learn the *exact same* set of underlying features, yet appear dissimilar under linear metrics like RSA and Linear Regression if they mix those features differently across neurons. While different feature arrangements may reflect genuine differences in how networks organize – and therefore act on – information, this phenomenon creates an unfair comparison problem: higher-dimensional models achieve higher alignment scores simply because they can represent features with less superposition (i.e., closer to one feature per neuron), making them inherently more linearly decodable (Elmoznino & Bonner, 2024). This dimensional advantage occurs even when comparing to lower-dimensional models with greater behavioral similarity to a target network.

We propose *feature-based alignment* to address these confounds and explore more performance-relevant sources of representational (mis)alignment. The key insight is that if superposition causes networks with identical features to appear misaligned, then *disentangling* those features should reveal their true similarity. Our approach has two steps: (1) extract disentangled features from each network's activations, and (2) compare networks using standard alignment metrics (RSA, Linear Regression) applied to these disentangled feature representations rather than raw neural activations. We fix the dimensionality of the disentangled space to be identical across all models, alleviating the dimensional advantage that confounds standard comparisons. For deep neural networks, we disentangle features using sparse autoencoders (SAEs) (Ng et al., 2011; Cunningham et al., 2023; Rao et al., 2024; Lan et al., 2024), a form of sparse dictionary learning (Olshausen & Field, 1997) that learns an overcomplete basis for neural activations. SAEs aim to represent each input as a sparse combination of interpretable features (Bricken et al., 2023), effectively reversing the feature mixing that occurs in superposition. For biological neural data (fMRI), where meaningful sparse features are more difficult to extract, we use K-means clustering on the mixed-selective neural responses instead.

In this work, we develop an analytic theory that quantifies how feature mixtures in superposition lead to misalignment under RSA and Linear Regression, and validate it across settings of increasing complexity. Applying feature-based alignment to real neural networks, we find that disentanglement often increases alignment between systems, but also observe cases where relative alignment between networks changes—with some networks becoming less similar in the latent space. Through simulations and analysis of feature representations, we identify that alignment increases with shared feature arrangements and feature weights. This is consistent with recent work showing elevated alignment with increased overlap in training data (which influences feature arrangements) and shared training

objectives (which influence feature weights and inductive biases) Li et al. (2025).Together, our results demonstrate that feature-based alignment facilitates fair comparisons and allows us to more directly observe the factors (i.e., feature arrangements and biases) that truly differentiate neural systems.

## 2 THEORY

Let $z \in \mathbb{R}^n$ be *latent variables* and $y \in \mathbb{R}^m$ be neural *representations*, which are functions of these latent variables, i.e., $y = f(z)$.

**Definition 2.1 (Superposition).** *We say that a representation $f : \mathbb{R}^n \to \mathbb{R}^m$ is in superposition if it is a linear map and a low-dimensional projection, i.e., $m < n$.*

### 2.1 ASSUMPTIONS

Throughout our analysis, we make the following assumptions:

1. **Linearity:** The neural representations are in superposition and are thus linear, described by a matrix $A \in \mathbb{R}^{m \times n}$:
$$y = Az \tag{1}$$
The condition $m < n$ implies that the columns of $A$ are not all orthogonal, aligning with the common assumption of having fewer neurons than latent variables.

2. **Sparsity of Latent Variables:** The latent variables are sparse, e.g., $\|z\|_0 \leq K$ for some $K \ll n$.

3. **Restricted Isometry Property (RIP):** The matrix $A$ satisfies the RIP, which allows for the theoretical possibility of recovering $z$ from observations of $y$ via compressed sensing.

4. **Distribution of Latent Variables:** For a dataset of $d$ inputs, the latent vectors $z_1, \ldots, z_d$ are treated as independent and identically distributed (i.i.d.) random variables satisfying:
   - Zero mean: $\mathbb{E}[z_i] = \mathbf{0}$ for all $i$.
   - White distribution (Identity covariance): $\mathbb{E}[z_i z_i^\mathsf{T}] = I_n$ for all $i$.

If these assumptions do not fully hold, we incur an irreducible reconstruction error when retrieving the sparse codes. This error would lower the ceiling of RSA alignment, correctly reflecting that if two features cannot be separated in one system, it should count as a representational misalignment.

### 2.2 REPRESENTATIONAL SIMILARITY MATRIX (RSM)

For a dataset of neural responses $Y = (y_1, ..., y_d)$, the *representational similarity matrix* (RSM) is defined as:
$$M(Y)_{i,j} = \langle y_i, y_j \rangle \quad \forall i, j \in \{1, ..., d\}. \tag{2}$$
Given the linearity assumption equation 1, we can rewrite the RSM in terms of the latent variables:
$$M(Y)_{i,j} = \langle y_i, y_j \rangle = \langle Az_i, Az_j \rangle = z_i^\mathsf{T} A^\mathsf{T} A z_j \tag{3}$$
This shows that the similarity between latent variables $z_i, z_j$ is measured by a semi-inner product $\langle \cdot, \cdot \rangle_G$ induced by the positive semi-definite Gram matrix $G := A^\mathsf{T} A$.

## 3 ALIGNMENT UNDER SUPERPOSITION

Consider two neural representations in superposition, with matrices $A_a, A_b$, generating responses $Y_a = (A_a z_1, ..., A_a z_d)$ and $Y_b = (A_b z_1, ..., A_b z_d)$ to the same set of latent variables $Z = (z_1, ..., z_d)$. While the underlying latent variables are identical, the observed neural representations $Y_a$ and $Y_b$ may differ. We now analyze how standard alignment metrics behave in this scenario.

The key insight of our work is that while these two neural representations $Y_a, Y_b$ originate from the same latent variables, any direct linear measure of alignment will be confounded by the differing projection matrices.

### 3.1 REPRESENTATIONAL SIMILARITY ANALYSIS (RSA)

The RSA metric is the Pearson correlation between the vectorized upper-triangular elements of two RSMs, $\vec{m}_a$ and $\vec{m}_b$.

$$\rho(Y_a, Y_b) = \frac{\text{Cov}(\vec{m}_a, \vec{m}_b)}{\sqrt{\text{Var}(\vec{m}_a)\text{Var}(\vec{m}_b)}} \tag{4}$$

Under the assumptions outlined previously, we arrive at the following result in the limit of large datasets.

**Theorem 3.1** (Asymptotic RSA Alignment). *The RSA correlation between two representations $Y_a$ and $Y_b$ in superposition is approximately the cosine similarity between their respective Gram matrices, $G_a = A_a^\mathsf{T} A_a$ and $G_b = A_b^\mathsf{T} A_b$.*

$$\boxed{\rho(Y_a, Y_b) \approx \frac{Tr(G_a G_b)}{\sqrt{Tr(G_a^2)Tr(G_b^2)}} = \frac{\langle G_a, G_b \rangle_F}{\|G_a\|_F \|G_b\|_F}} \tag{5}$$

*where $\langle \cdot, \cdot \rangle_F$ and $\| \cdot \|_F$ are the Frobenius inner product and norm, respectively.*

This result shows that RSA is fundamentally sensitive to the similarity of the metric tensors induced by the representations on the latent space.

### 3.2 LINEAR REGRESSION

Alternatively, we can measure alignment by determining how well one representation can be linearly predicted from the other using a multivariate linear model $Y_b = W Y_a + E$. The Ordinary Least Squares (OLS) estimator $\hat{W}$ minimizes the squared Frobenius norm of the residuals, $\|Y_b - W Y_a\|_F^2$.

**Theorem 3.2** (Asymptotic Linear Regression). *In the asymptotic limit and under the stated assumptions, the OLS estimator $\hat{W}$ and the resulting model performance are given by:*

1. ***Optimal Weights:*** *The weight matrix $\hat{W}$ converges to:*

$$\boxed{\hat{W} \approx A_b A_a^\mathsf{T} (A_a A_a^\mathsf{T})^{-1}} \tag{6}$$

2. ***Mean-Squared Error (MSE):***

$$\boxed{MSE(Y_b | Y_a) \approx \frac{1}{m_b} \left\| A_b - \hat{W} A_a \right\|_F^2} \tag{7}$$

3. ***Explained Variance ($R^2$):***

$$\boxed{R^2 = 1 - \frac{Tr\left((A_b - \hat{W} A_a)^\mathsf{T}(A_b - \hat{W} A_a)\right)}{Tr(A_b^\mathsf{T} A_b)}} \tag{8}$$

4. ***Pearson Correlation ($\rho(\hat{Y}_b, Y_b)_{ij}$):***

$$\boxed{\rho(\hat{Y}_b, Y_b)_{ij} = \frac{(\hat{W} A_a A_b^\mathsf{T})_{ij}}{\sqrt{(\hat{W} A_a A_b^\mathsf{T})_{ii}(A_b A_b^\mathsf{T})_{jj}}}} \tag{9}$$

## 4 SUPERPOSITION'S IMPACT ON ALIGNMENT IN REAL NETWORKS

### 4.1 EXPERIMENTAL SETUP

After verifying that idiosyncratic superposition arrangements are sufficient to reduce alignment (Fig 7), we now test whether superposition disentanglement changes alignment in real neural networks.

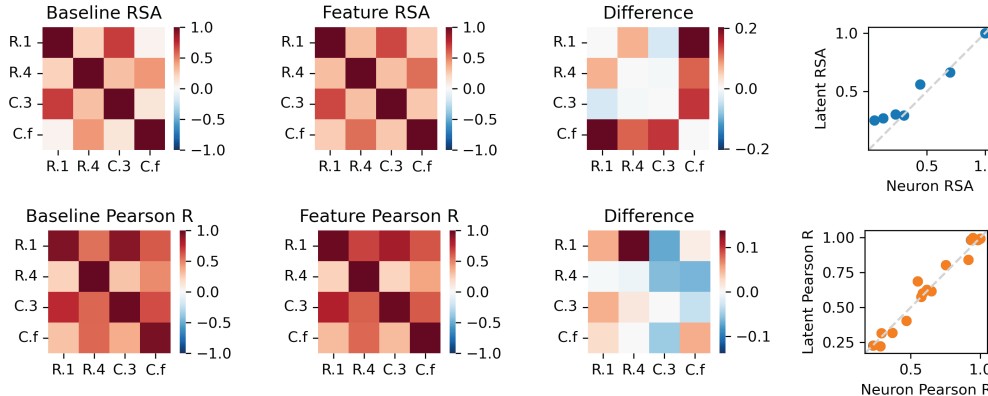

Figure 2: **Model-Model Comparison for SAE latents.** **Top Plots:** Heatmaps: Neuron based RSA (left), latent based RSA (middle), and difference (right). Scatterplot: Neuron versus latent based RSA. **Bottom Plots:** Same as top row, but for Linear Regression. On the scatterplot, blue datapoints indicate the X axis was used as the source for Linear Regression mapping, and orange points indicate the Y axis was used as the source for Linear Regression mapping.

We measure model-model (Fig. 2), model-brain (Fig. 3), and brain-brain (Fig. 4) alignment using RSA and Linear Regression. To begin, we measure alignment on raw neural activations to obtain a baseline. Next, we train SAEs and K-Means on models and brains to recover latent features and use them in place of neurons for computing alignment. For RSA, we replace the neurons of both systems with latents, whereas with Linear Regression, only the source neurons are replaced with latents. This is done to keep the targets the same as in the base comparison (i.e., predicting neurons). It is technically sound because Linear Regression is capable of remixing the source latents back into the target's superposition arrangement. Finally, we report the difference between alignment over latent activations and alignment over raw neural activations to quantify the relative increase in alignment provided by disentangling features from superposition.

## 4.2 DATA

We obtained neural activations from both biological and artificial neural networks. Biological data is from the publicly available Natural Scenes Dataset (NSD) (Allen et al., 2022), which uses fMRI to record human neural responses to subsets of the COCO natural images dataset (Lin et al., 2014). We use data from six brain areas along the visual processing hierarchy: early to mid-level visual cortex (V1v, V2v, V3v, hV4), the occipital face area (OFA) and the fusiform face area 1 (FFA-1). All activations were preprocessed (the result of Step 5 described in (Allen et al., 2022)) neural responses from NSD Subject 1 in response to 10,000 unique COCO images. Each neural response was averaged over 3 image presentations and z-scored.

Model activations are from the early and penultimate layers of ResNet-50 (layer 1 and layer4.2) (He et al., 2016) and CLIP-ViT-B/32 (layer 3 and feature layers) (Radford et al., 2021). Both models are trained on ImageNet classification (Deng et al., 2009), with activations from the same 10,000 images viewed by Subject 1 of the NSD for consistency.

## 4.3 SAE TRAINING

We train sparse autoencoders with an L1 sparsity penalty (L1-SAEs) to learn disentangled latent features ($z$). The SAE has an encoder and a decoder. Encoding is given by:

$$z = \text{ReLU}(W_{\text{enc}}x + b_{\text{enc}})$$

where $x$ represents the raw neural activations, and learned parameters $W_{\text{enc}}$ and $b_{\text{enc}}$ are the encoder weights and bias respectively. Decoding is given by:

$$\hat{x} = W_{\text{dec}}z + b_{\text{dec}}$$

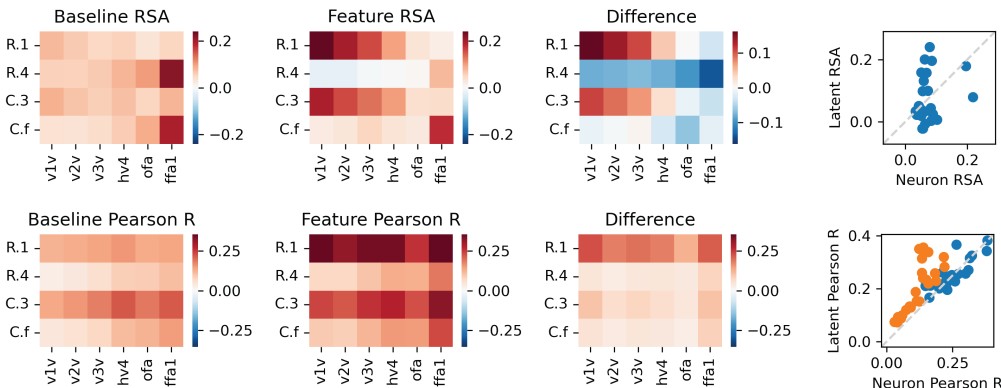

Figure 3: **Model-Brain Comparison for K-Means latents. Top Plots:** Heatmaps: Neuron based RSA (left), latent based RSA (middle), and difference (right). Scatterplot: Neuron versus latent based RSA. **Bottom Plots:** Same as top row, but for Linear Regression. On the scatterplot, blue datapoints indicate the X axis was used as the source for Linear Regression mapping, and orange points indicate the Y axis was used as the source for Linear Regression mapping.

where $\hat{x}$ are reconstructed neural activations, and learned parameters $W_{\text{dec}}$ and $b_{\text{dec}}$ are the decoder weights and bias respectively. The model is trained using a combined loss function, which is the sum of a reconstruction loss

$$\mathcal{L}_{\text{reconstruction}} = \frac{1}{d \cdot M} \sum_{i=1}^{d} (x_i - \hat{x}_i)^2$$

and sparsity loss

$$\mathcal{L}_{\text{sparsity}} = \frac{\lambda}{d \cdot N} \sum_{i=1}^{d} \sum_{j=1}^{N} |(W_{\text{dec}})_{:,j}| \cdot |z_i^j|$$

which is the L1 norm of latent activations scaled by the decoder norm (to avoid collapse with vanishing latents and exploding decoder norms) and weighted by the hyperparameter $\lambda$. We varied $\lambda$ from $10^{-3}$ to 20 and set the number of latent dimensions to 2048 for all neural networks.

We train SAEs on activations of all models and brains to the the 10,000 Natural Scenes Dataset (NSD) images shown to Subject 1 in the Allen et al. (2022) study. A total of 100 SAEs are trained on each set of neural responses. We choose the best SAE using an unsupervised metric described in section 4.5

## 4.4 K-MEANS LATENT TRANSFORMATION

We perform K-means clustering over columns (images) on the original (MxI) neural datasets, where M is the number of neurons and I is the number of images. In the resulting feature space of N clusters, each cluster represents a visual feature (i.e., cat images), each datapoint is an (M, 1) vector containing all single-neuron responses to one image, and each centroid can be thought of as representing the canonical population response associated with a particular visual feature. We transform each original datapoint (M, 1) into a population response vector (N, 1) by computing the negative Euclidean distance between the datapoint and N cluster centers. This results in a population response dataset (NxI), which represents the distance of each population vector from the canonical response to a given feature. We train 50 randomly initialized K-means seeds per neural network comparison, choosing the best model with an unsupervised metric outlined in section 4.5.

## 4.5 MODEL SELECTION AND VALIDATION

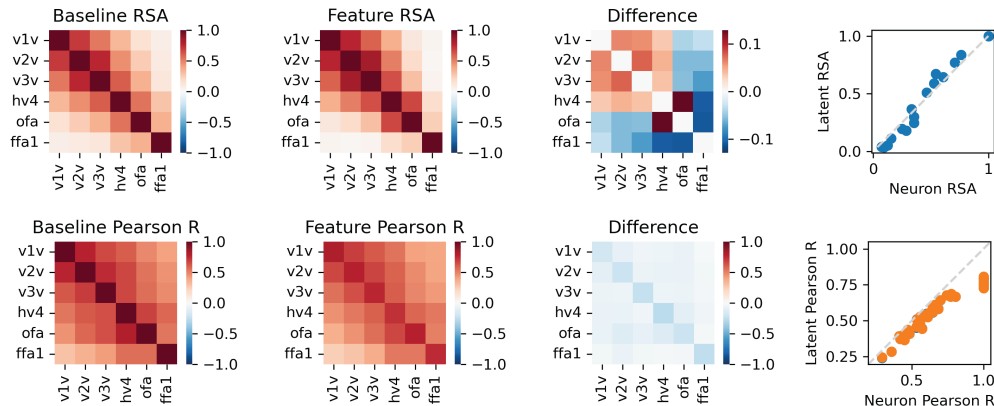

Figure 4: **(Within-Subject) Brain-Brain Comparison for K-Means latents.** **Top Plots:** Heatmaps: Neuron based RSA (left), latent based RSA (middle), and difference (right). Scatterplot: Neuron versus latent based RSA. **Bottom Plots:** Same as top row, but for Linear Regression. On the scatterplot, blue datapoints indicate the X axis was used as the source for Linear Regression mapping, and orange points indicate the Y axis was used as the source for Linear Regression mapping.

For both SAEs and K-Means, we report alignment results using the most disentangled model, identified via a variant of the Unsupervised Disentanglement Ranking (UDR) metric (Higgins et al., 2021). Briefly, we train multiple models (100 for SAEs, 50 for K-Means) and compute an RSA-based similarity matrix across all models. The model with the highest average pairwise similarity to all others receives the highest UDR score and is considered most disentangled, as it represents the most consistent solution across the optimization landscape. Validating this approach, we find that UDR scores correlate with alignment performance: models with higher UDR achieve higher cross-system alignment (Figure 8).

To verify that UDR-selected models produce visually interpretable features consistent with observed alignment changes, we employed an automated interpretability metric derived from human psychophysics. This metric quantifies feature interpretability through an odd-one-out task, analogous to word intrusion tasks used to evaluate topic models Chang et al. (2009), but adapted for the visual modality. We identify the top K preferred images (or maximally exciting images; MEIs) for each neuron or latent and compute their average pairwise similarity to establish a top K image similarity threshold. We then compute the average similarity between each remaining image in the dataset and these top K images. The feature or neuron receives one point for each image whose average similarity falls below the top K threshold, indicating the algorithm correctly identified it as an 'odd one out' or dissimilar to the feature's preferred stimuli. Higher odd-one-out scores indicate more interpretable features with consistent selectivity. We visualize the preferred images of the most interpretable features for a subset of comparisons in the Appendix.

### 4.6 RESULTS

**Model to Model.** Alignment results between models are presented in Figure 2. **RSA:** Both neural and feature space showed the highest similarity between more analogous model layers. Feature-based alignment yielded overall higher scores. Notably, ResNet-50 layer 1 showed a shift in its alignment profile, with the highest alignment increase with the CLIP feature layer, followed by ResNet50-layer 4 and decreased alignment with earlier CLIP layer 3. Figure 9 visualizes the preferred images for neurons versus latents, confirming greater correspondence in preferred features between the CLIP feature layer and ResNet-50 layer 1 in latent space compared to neural space. **Linear Regression:** As with RSA, early model layers are most related. Unlike RSA, late model layers showed less selective similarity profiles, and feature-based alignment did not produce a pronounced overall increase in alignment scores. Where alignment increased in RSA, it sometimes decreased with Linear Regression (e.g., CLIP feature layer's comparisons to both ResNet50 layer3 and CLIP layer 3), and this was enough to invert similarity profiles in feature space for Linear Re-

gression relative to RSA (e.g., for the CLIP feature layer). The opposite was also true: ResNet-50 layer 1 and CLIP layer 3 become more similar with feature-based Linear Regression, and less similar with feature-based RSA. We explore the sources metric disagreement in Section 5.

**Model to Brain.**   Alignment results for model-to-brain comparisons are presented in Figure 3. **RSA:** In neural space, early model layers roughly aligned more strongly with early visual cortex (V1-V3) while later layers aligned with late visual cortex (V4-FFA-1). Feature-based alignment strengthened this hierarchical bias for early layers/regions and decreased it for later layers/regions. **Linear Regression:** In neural space, early model layers showed broader alignment across visual cortex, with a subtle hierarchical alignment observed for later model layers. Feature-based alignment produced different effects than RSA. Early layers, particularly ResNet-50 layer 1, became more strongly aligned to all visual cortical regions. Late layers showed modest increases in alignment—contrasting with the decreases observed using RSA. We address potential causes of the differences between Linear Regression and RSA in Section 5.

**Brain to Brain.**   Alignment results for brain-to-brain comparisons are presented in Figure 4. **RSA:** Both neural and feature space exhibited hierarchically organized alignment, with neighboring visual regions showing greater similarity. Feature-based alignment strengthened this pattern for early visual areas but weakened it for higher-order regions. Notably, hV4—a mid-level visual region—shifted its alignment profile in feature space: while most similar to V3v in neural space, it became most similar to OFA (a face-selective area) in feature space. Figure 10 visualizes the preferred images for neurons versus latents in V3v, hV4, and OFA, demonstrating greater overlap in preferred features between hV4 and OFA in the latent space compared to the neural space. This shift in relative alignment demonstrates how feature-based methods can reveal functional relationships obscured by neural-level comparisons. We hypothesize this shift arises because disentanglement reduces the geometric effects of OFA's strong bias towards facial features, allowing shared mid-level representations to emerge. We explore this mechanism in Section 5. **Linear Regression:** Neural space showed weaker hierarchical organization than RSA, though neighboring regions still exhibited some preferential alignment. In contrast to RSA, feature-based alignment uniformly decreased similarity scores across all region pairs, suggesting that Linear Regression is differentially sensitive to disentanglement. We elaborate on how bias may also contribute to RSA - Linear Regression disagreement in Section 5.

## 5   INVESTIGATING SOURCES OF REPRESENTATIONAL ALIGNMENT

In the previous section, we observed several intriguing trends in neural alignment. First, early and lower-dimensional model layers and brain regions exhibited increased alignment in feature space for both RSA and Linear Regression, consistent with our initial hypothesis about superposition arrangements obscuring their true similarity. Second, higher-order brain regions with similar intrinsic dimensionality to lower-level areas often exhibited decreases in alignment. In several of these cases, RSA and Linear Regression even disagreed, causing changes in selectivity profiles for regions measured with one metric but not the other. These last two findings prompt us to investigate whether (un)known inductive biases (e.g., shared face selectivity), particularly of higher-order regions, contributes to relatively high alignment in the neural space that is reduced in feature space.

### 5.1   EXPERIMENTAL SETUP FOR SIMULATION STUDIES

For all simulation studies, we produce two random linear projections from a shared set of features. Specifically, we generate a single feature set $Z$ of $d \times N$ dimensional features (i.e. $Z \in \mathbb{R}^{d \times N}$), which are random uniform values between 0 and 1, i.e., $Z_{i,j} \sim \mathcal{U}(0, 1)$. To simulate the sparsity condition, we then zero mask all but the top $K$ activating latent variables within each generated sample (i.e. individual row in $Z$). Next, we generate two projection matrices, each $N \times M$ dimensional, with elements drawn from a standard normal distribution, i.e., $A_0, A_1 \in \mathbb{R}^{N \times M}$ where $A_{i,j} \sim \mathcal{N}(0, 1)$. These matrices are used to produce two random linear projections of a shared set of features. We perform Linear Regression and RSA on the resulting simulated neural activations.

In Experiment 1, we simulate the impact of shared feature arrangements by progressively constraining the projection matrices $A_0$ and $A_1$ such that features maintain similar projection patterns across systems. This is achieved by generating a random feature correlation matrix and multiplying it with an increasing number of columns in $A_0$ and $A_1$. In Experiment 2, we simulate the impact of shared biases by multiplying the columns of projection matrices $A_0$ and $A_1$ with progressively larger weights from a feature importance matrix. This matrix follows an exponential decay function that assigns the highest weights to the initial features. In Experiment 3, we simulate the impact of dimensionality on one of the networks by increasing its dimensionality through a scalar multiplier.

## 5.2 SIMULATION STUDY RESULTS

Experiment 1 reveals that overlapping feature arrangements increase alignment similarly for both RSA and Linear Regression, consistent with shared training statistics benefiting both metrics. Experiment 2 shows a strong dissociation: shared feature bias decreases Linear Regression alignment but increases RSA alignment, with RSA yielding higher absolute scores when bias is sufficiently strong. This mirrors the metric dissociations we observed in real neural data for multiple comparisons involving FFA-1, a region known to exhibit bias towards faces. Experiment 3 demonstrates that the RSA-Linear Regression gap is amplified by high dimensionality, confirming that dimensional mismatches disproportionately inflate Linear Regression scores.

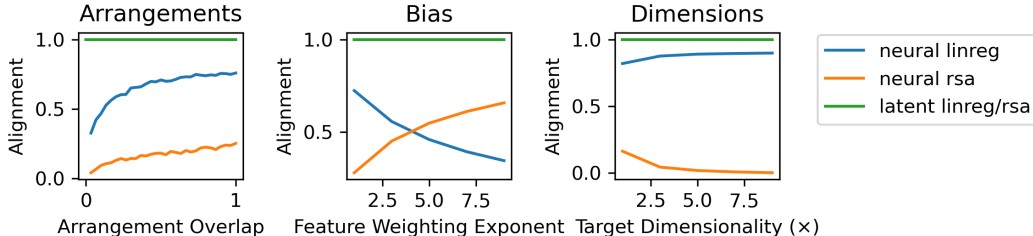

Figure 5: **Sources of (mis) alignment in neural space Left:** Simulation manipulating the degree of shared feature arrangement statistics (Experiment 1). **Middle:** Simulation manipulating the strength of shared bias (Experiment 2). **Right:** Simulation manipulating the dimensionality of the target neural network (Experiment 3).

## 5.3 EXTENSION TO REAL DATA

Experiment 2 of the previous section (simulating bias) represents the only condition where RSA yields higher alignment than Linear Regression and where the same manipulation produces opposing directional effects on the two metrics. As this means RSA-Linear Regression disagreement of this nature might be a diagnostic indicator for bias, we focus our analysis on the real data in this section on bias. We identified cases where RSA > Linear Regression in the neural space: ResNet-50 layer 4 to FFA-1 and CLIP feature layer to FFA-1. We sort neural activity for each system according to the L-1 norm to identify the top 10 features for each system, and found they overlap in their semantic selectivity more than features where RSA <= Linear Regression 6. We apply the same L1-sorting strategy to the latents of each system, finding a decrease in semantic selectivity over the top 10 features that coincides with the decrease in RSA observed in feature-based alignment. Visual inspection confirms the nature of this shared bias: Figure 6. All systems in this comparison show strong selectivity for faces and human figures—a well-documented inductive bias in both deep networks and FFA-1. This concentration of shared semantic selectivity in high-magnitude features indicates that high RSA, coupled with RSA-Linear Regression dissociation, may be diagnostic of shared feature-level biases.

# 6 LIMITATIONS

There are several limitations in our study. The first is our assumptions that 1) projections from the latent to neural basis are random and 2) that all features are shared. These assumptions are purely

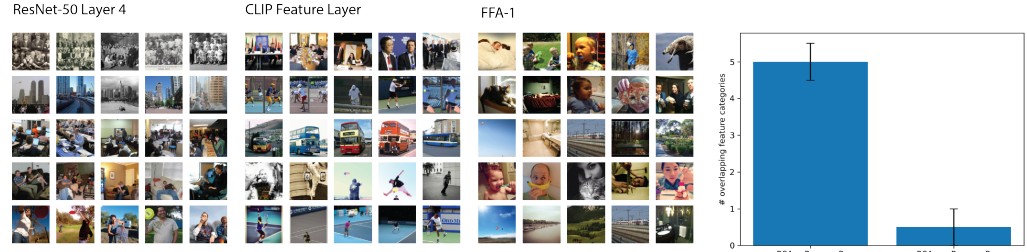

Figure 6: **Preferred features of neural networks with high baseline RSA**. **Images:** Top 5 Maximally Exciting Images (MEIs) for the top 5 features from ResNet-50 Layer 4 (Left), the CLIP feature layer (Middle) and FFA-1 (Right). **Barplot:** Degree of categorical overlap for the top 10 MEIs for high baseline RSA vs low baseline RSA comparisons.

practical; allowing us to test whether disentangling superimposed features is sufficient to increase true alignment in cases where feature arrangements obscure it. At certain scales and in certain areas, biological neural networks have a bias towards privileged, rather than random, projections (Khosla et al., 2024; Posani et al., 2025). The impact of this on alignment is likely complex and worth further exploration. It is also unlikely that all of the real networks in our study represent the exact same feature set. The second limitation stems from our use of SAEs, known to suffer various problems such as an amortization gap O'Neill et al. (2024), inconsistent latents across training seeds (Paulo & Belrose, 2025) and the sensitivity of discovered latents to dictionary dimensionality (Leask et al., 2025; Chanin et al., 2024). Further work could explore recent efforts to alleviate such problems (Fel et al., 2025), but we stress that our theory does not depend on SAEs. We pragmatically adopt SAEs as the current best method to disentangle features in superposition, and our experiments should be revisited if improved approaches are designed. The final limitation concerns scope: we only test linearly combined features. This is grounded in the superposition hypothesis (Elhage et al., 2022) and the success of linear and SAE-based probing in large models, which demonstrate that many features are linearly combined and linearly recoverable. However, the success of nonlinear metrics Huh et al. (2024); Insulla et al. (2025); Kornblith et al. (2019); Williams et al. (2021) suggests that follow-up studies may uncover additional sources of alignment obscured in neural space.

## 7 DISCUSSION

In this work, we derive analytic predictions and contribute simulation experiments demonstrating that representational alignment decreases as a function of distinct superposition arrangements of the same underlying features (i.e., compression via random projections). These experiments suggested that alignment computed over disentangled features would be higher. Based on this prediction, we used SAEs and K-Means to extract approximations of features in real neural networks, showing that alignment over latent activations is often significantly higher for the commonly used metrics of RSA and Linear Regression, particularly for early, lower-dimensional layers. We also observe a restructuring of relative representational similarities between models and across biological and artificial networks. Our findings have implications for model selection criteria. If superposition masks similarity between two systems that represent even identical features, then computing RSA or Linear Regression over raw activations of models with variable dimensionality places smaller models at a systematic disadvantage. This may explain why scaling models often produces more reliable alignment gains than designing models with more apparent alignment to human perception (Schaeffer et al., 2022; 2024). Additionally, identifying the causes of restructured representational similarity in feature space may help explain why two systems are similar in neural space, and whether this stems from a dimensionality confound or a more substantive property of the neural networks (i.e., shared inductive biases). As we seek to understand whether models and brains share representational strategies, it is important to consider the best uses of common alignment metrics. In this work, we demonstrate that performing alignment on raw neural activations imposes a systematic disadvantage for earlier, lower-dimensional models. We offer superposition disentanglement as a practical and effective solution to address this confound currently facing neural network comparisons with otherwise similar behavior.

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

# A    APPENDIX

# B    SIMULATING SUPERPOSITION'S IMPACT ON ALIGNMENT

## B.1    EXPERIMENTAL SETUP

In this section, we test our theoretical prediction that superposition is sufficient to reduce alignment in cases where two networks use an identical set of features. We generate a single feature set $Z$ of $d \times N$ dimensional features (i.e. $Z \in \mathbb{R}^{d \times N}$), which are random uniform values between 0 and 1, i.e., $Z_{i,j} \sim \mathcal{U}(0, 1)$. To simulate the sparsity condition, we then zero mask all but the top $K$ activating latent variables within each generated sample (i.e. individual row in $Z$). Next, we generate two projection matrices, each $N \times M$ dimensional, with elements drawn from a standard normal distribution, i.e., $A_0, A_1 \in \mathbb{R}^{N \times M}$ where $A_{i,j} \sim \mathcal{N}(0, 1)$. These matrices are used to produce two random linear projections of a shared set of features. We manipulate the degrees of superposition by varying $M$ from $0.2K \log(N/K)$ to $50K \log(N/K)$. Next, we measured alignment of the random linear projections using RSA (Experiment 1) and Linear Regression (Experiment 2). To test the effect of sparsity, we repeated these experiments across different numbers of active latents ($K$). We also calculate and show the minimum dimensionality of $M$ required for accurate latent recovery under compressed sensing as $M = K \log(N/K)$ (Candes et al., 2006).

## B.2    RESULTS

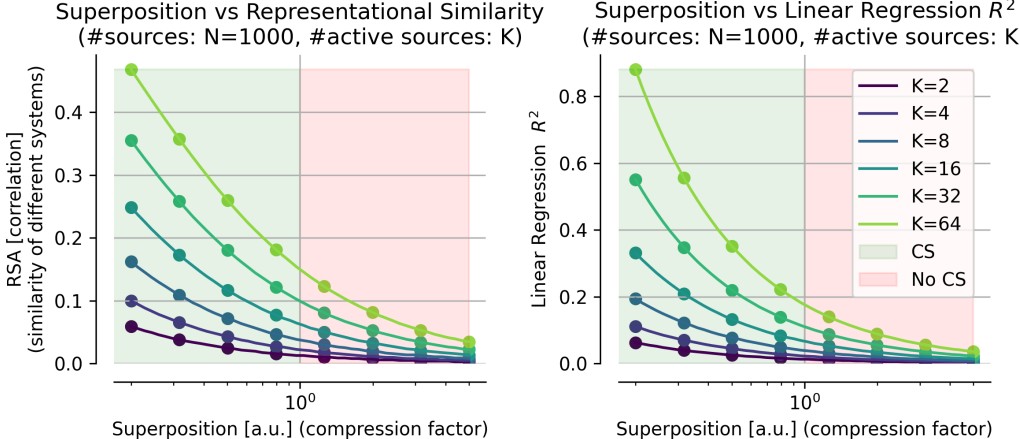

Figure 7: **Neural Network Alignment Decreases with Superposition.** Alignment measured with RSA (**Left**) as well as with Linear Regression (**Right**) as a function of compression ($N/M$). This experiment is repeated across multiple sparsity levels ($K$). Analytical predictions are represented by solid curves, while empirical results from simulation across different superposition compressions is represented by the dots. We note where accurate latent recovery from compressed representations is (CS; green shading) or is not (No CS; red shading) possible Donoho (2006).

## B.3    DERIVATION OF ANALYTICAL RSA

To derive an analytic expression for the RSA under superposition, we first express the RSMs in terms of the Gram matrices $G_a = A_a^\mathsf{T} A_a$ and $G_b = A_b^\mathsf{T} A_b$. These matrices act as metric tensors, defining the geometry of the representations.

$$M(Y_a) = (A_a Z)^\mathsf{T}(A_a Z) = Z^\mathsf{T} G_a Z \tag{10}$$

$$M(Y_b) = (A_b Z)^\mathsf{T}(A_b Z) = Z^\mathsf{T} G_b Z \tag{11}$$

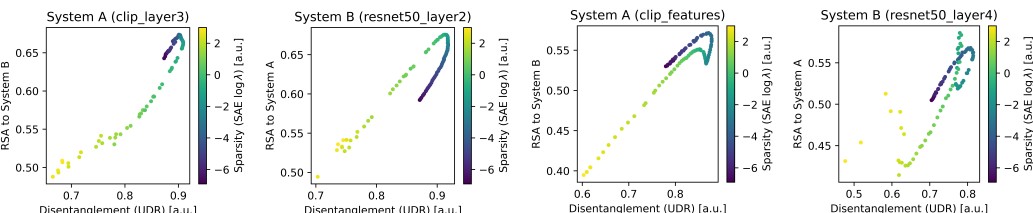

Figure 8: **A** We plot the UDR of trained SAEs for clip_layer3 (system A) against alignment of those SAEs with resnet50_layer2 (system B), finding that high UDR scores coincide with high alignment. **B** We perform the reverse comparison to select the most disentangled model for resnet50_layer2.**C-D** Same as A-B, but selecting models for clip_features (C) and resnet50-layer4 (D).

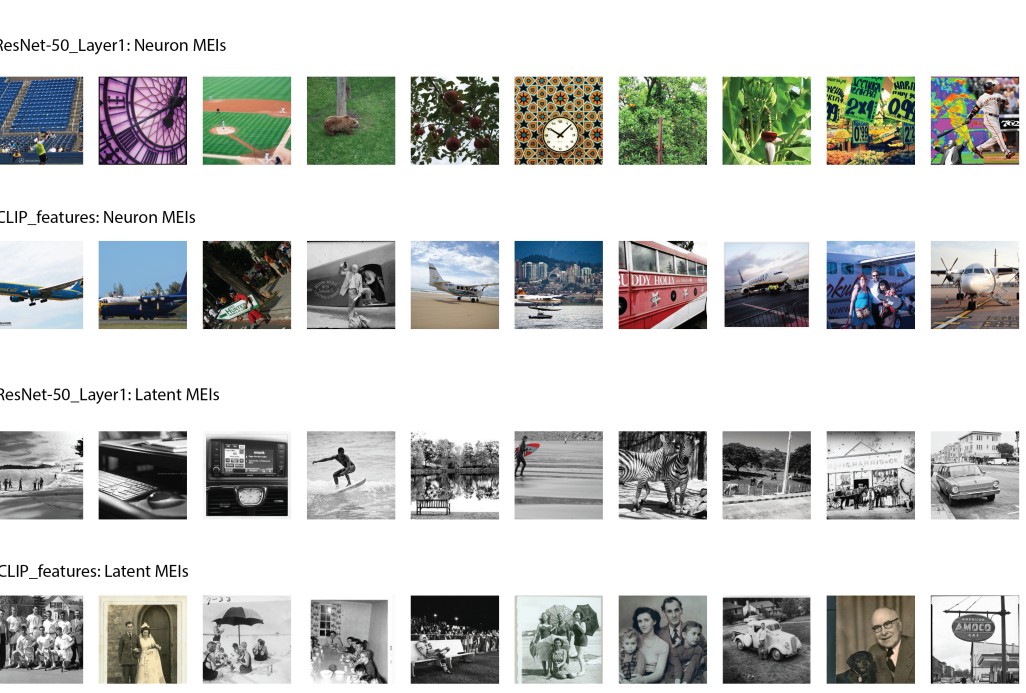

Figure 9: **SAE latents MEIs for Model-Model Comparisons. Top rows:** 10 maximally exciting images (MEIs) for the most interpretable neuron from ResNet-50 Layer 1 and CLIP feature layer. **Bottom rows:** 10 maximally exciting images (MEIs) for the most interpretable latent from ResNet-50 Layer 1 and CLIP feature layer. This supports the increase in feature-based alignment between ResNet-50 Layer 1 and the CLIP feature layer observed in Figure 3.

An individual element of these matrices is the quadratic form $M(Y_a)_{ij} = z_i^\mathsf{T} G_a z_j$. Our derivation relies on the following standard assumptions about the distribution of the latent variable vectors $z_i$:

1. The latent vectors $z_1, \ldots, z_d$ are independent and identically distributed (i.i.d.).

2. The distribution has a mean of zero: $\mathbb{E}[z_i] = \mathbf{0}$.

3. The distribution is white, with an identity covariance matrix: $\mathbb{E}[z_i z_j^T] = \delta_{ij} I_n$.

**Expectation of RSM Elements**  We first derive the empirical mean of all RSM matrix elements $\mu_Y$ in asymptotic limit, then derive the empirical mean of only the off-diagonal upper triangular

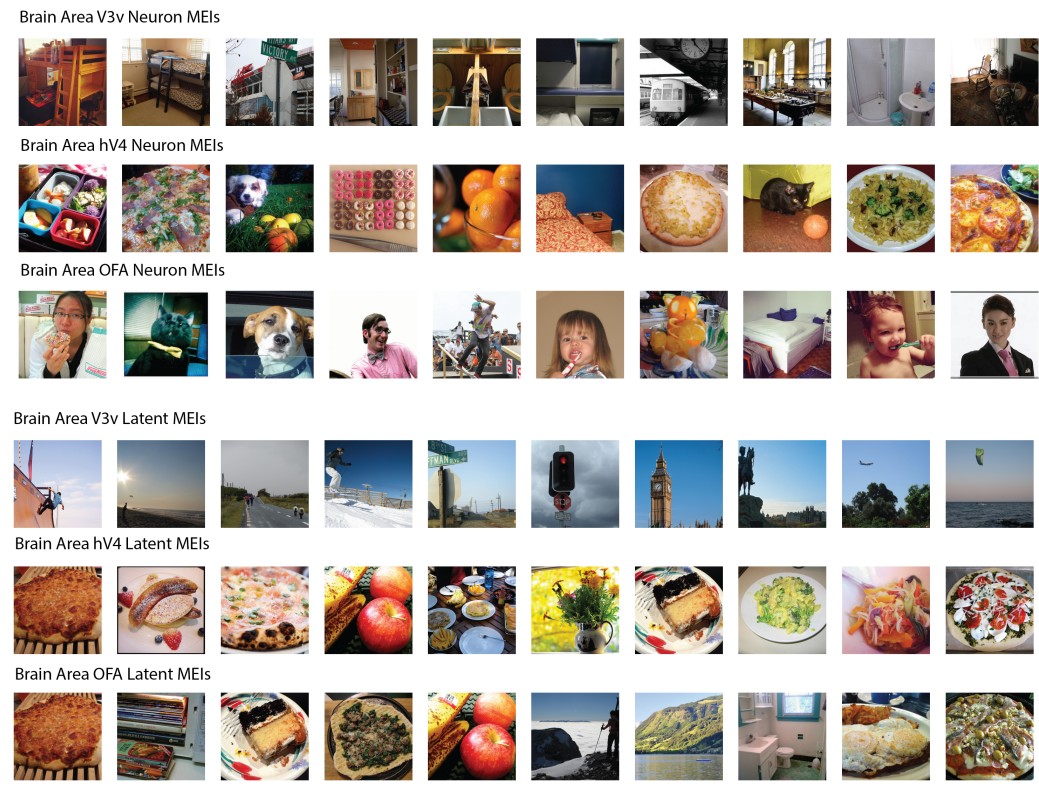

Figure 10: **K-means latents MEIs for Brain-Brain Comparisons.** **Top rows:** 10 maximally exciting images (MEIs) for the most interpretable neuron from brain areas V3v, hV4 and OFA. **Bottom rows:** 10 maximally exciting images (MEIs) for the most interpretable latent from brain areas V3v, hV4 and OFA. This supports the switch from higher hV4-V3v similarity in the neural space to higher hV4-OFA similarity in the latent space observed in Figure 5.

RSM matrix elements $\mu_Y^{UT}$, and show that in the asymptotic limit the two empirical quantities are equivalent and converge to zero:

$$\mu_Y \equiv \frac{1}{d^2} \sum_{i,j} M(Y)_{ij} = \frac{1}{d^2} \sum_{i,j} z_i^T G z_j \tag{12}$$

$$= \frac{1}{d} \sum_i z_i^T G \left[ \frac{1}{d} \sum_j z_j \right] \tag{13}$$

$$\approx \frac{1}{d} \sum_i z_i^T G \mathbb{E}[z_j] = z_i^T G \mathbf{0} \tag{14}$$

$$= 0 \tag{15}$$

$$\mu_Y^{UT} \equiv \frac{1}{d(d-1)/2} \sum_{i<j} M(Y)_{ij} = \frac{1}{d(d-1)} \sum_{i \neq j} M(Y)_{ij} \tag{16}$$

$$= \frac{1}{d(d-1)} \left\{ \left[ \sum_{i,j} M(Y)_{ij} \right] - \left[ \sum_i M(Y)_{ii} \right] \right\} \tag{17}$$

$$= \frac{d^2}{d(d-1)} \mu_Y - \frac{1}{d-1} \mu_Y^{\mathrm{diag}} \tag{18}$$

$$\approx \mu_Y \tag{19}$$

$$= 0 \tag{20}$$

**Covariance and Variance**  Since the mean of the off-diagonal elements is zero, their covariance for $i \neq j$ is the empirical mean of their product: The Covariance of the off-diagonal elements of two RSMs can then be shown as:

$$\mathrm{Cov}(\vec{m}_a, \vec{m}_b) = \mathrm{Cov}(M(Y_a)^{\mathrm{UT}}, M(Y_b)^{\mathrm{UT}}) \tag{21}$$

$$= \frac{1}{d(d-1)/2} \sum_{i<j} \{ M(Y_a)_{ij} - \mu_a^{\mathrm{UT}} \} \{ M(Y_b)_{ij} - \mu_b^{\mathrm{UT}} \} \tag{22}$$

$$\approx \frac{1}{d(d-1)/2} \sum_{i<j} M(Y_a)_{ij} M(Y_b)_{ij} = \frac{1}{d(d-1)} \sum_{i \neq j} M(Y_a)_{ij} M(Y_b)_{ij} \tag{23}$$

$$= \frac{1}{d(d-1)} \left\{ \left[ \sum_{i,j} M(Y_a)_{ij} M(Y_b)_{ij} \right] - \left[ \sum_i M(Y_a)_{ii} M(Y_b)_{ii} \right] \right\} \tag{24}$$

$$\approx \frac{1}{d(d-1)} \sum_{i,j} M(Y_a)_{ij} M(Y_b)_{ij} \tag{25}$$

$$= \frac{1}{d(d-1)} \sum_{i,j} (z_i^\mathsf{T} G_a z_j)(z_i^T G_b z_j) \tag{26}$$

$$= \frac{1}{d(d-1)} \sum_{i,j} (z_i^\mathsf{T} G_a z_j)(z_j^T G_b^\mathsf{T} z_i) \tag{27}$$

$$= \frac{1}{d-1} \sum_i z_i^\mathsf{T} G_a \left[ \frac{1}{d} \sum_j z_j z_j^\mathsf{T} \right] G_b z_i \tag{28}$$

$$\approx \frac{1}{d-1} \sum_i z_i^\mathsf{T} G_a \mathbb{E}[z_j z_j^\mathsf{T}] G_b z_i \tag{29}$$

$$= \frac{1}{d-1} \sum_i z_i^T G_a G_b z_i \tag{30}$$

$$= \frac{d}{d-1} \mathrm{Tr} \left[ G_a G_b \left( \frac{1}{d} \sum_i z_i z_i^\mathsf{T} \right) \right] \tag{31}$$

$$\approx \mathrm{Tr} \left[ G_a G_b \mathbb{E}[z z^\mathsf{T}] \right] \tag{32}$$

$$= \mathrm{Tr} \left[ G_a G_b \right] \tag{33}$$

The variance of the elements is found by setting $G_a = G_b$, and can be related to the Frobenius norm ($\|X\|_F^2 = \mathrm{Tr}(X^T X)$):

$$\mathrm{Var}(\vec{m}_a) = \mathrm{Var}(M(Y_a)^{\mathrm{UT}}) = \mathrm{Tr}(G_a G_a) = \mathrm{Tr}(G_a^\mathsf{T} G_a) = \|G_a\|_F^2 \tag{34}$$

$$\mathrm{Var}(\vec{m}_b) = \mathrm{Var}(M(Y_b)^{\mathrm{UT}}) = \mathrm{Tr}(G_b G_b) = \mathrm{Tr}(G_b^\mathsf{T} G_b) = \|G_b\|_F^2 \tag{35}$$

For a large number of data points $d$, the correlation of the vectorized RSMs is well-approximated by the correlation of their constituent elements. Substituting the covariance and variance into the

Pearson formula yields our main result:

$$\rho(Y_a, Y_b) \approx \frac{\text{Tr}(G_a G_b)}{\sqrt{\|G_a\|_F^2 \|G_b\|_F^2}} = \frac{\langle G_a, G_b \rangle_F}{\|G_a\|_F \|G_b\|_F} \qquad (36)$$

### B.4 DERIVATION OF ANALYTICAL LINEAR REGRESSION RESULTS

We consider a multivariate linear regression model to predict the activity of representation $Y_b$ from $Y_a$:

$$Y_b = WY_a + E \qquad (37)$$

where $W \in \mathbb{R}^{m_b \times m_a}$ is the weight matrix and $E$ is the matrix of residuals. The Ordinary Least Squares (OLS) method finds the estimator $\hat{W}$ that minimizes the sum of squared errors, given by the squared Frobenius norm $\|Y_b - WY_a\|_F^2$.

**OLS Estimator and Asymptotic Simplification** The standard OLS solution for the weight matrix is:

$$\hat{W} = Y_b Y_a^\mathsf{T} (Y_a Y_a^\mathsf{T})^{-1} \qquad (38)$$

To find an analytic expression in terms of the underlying superposition matrices, we substitute $Y_a = A_a Z$ and $Y_b = A_b Z$. We then leverage the same statistical properties of the latent variables $Z$ used in the RSA derivation. For a large number of i.i.d. samples $d$, the sample covariance of the latent variables converges to a scaled identity matrix:

$$\frac{1}{d} Z Z^\mathsf{T} = \frac{1}{d} \sum_{i=1}^{d} z_i z_i^\mathsf{T} \to \mathbb{E}[zz^\mathsf{T}] = I_n \quad \implies \quad Z Z^\mathsf{T} \approx d I_n$$

Using this approximation, the terms in the OLS estimator simplify:

$$Y_b Y_a^\mathsf{T} = (A_b Z)(A_a Z)^\mathsf{T} = A_b (Z Z^\mathsf{T}) A_a^\mathsf{T} \approx d(A_b A_a^\mathsf{T}) \qquad (39)$$

$$Y_a Y_a^\mathsf{T} = (A_a Z)(A_a Z)^\mathsf{T} = A_a (Z Z^\mathsf{T}) A_a^\mathsf{T} \approx d(A_a A_a^\mathsf{T}) \qquad (40)$$

Substituting these into the formula for $\hat{W}$ gives the ideal "population" level regression coefficient, which is free from the sampling noise of a specific $Z$:

$$\hat{W} \approx d(A_b A_a^\mathsf{T}) \left( d(A_a A_a^\mathsf{T}) \right)^{-1} = A_b A_a^\mathsf{T} (A_a A_a^\mathsf{T})^{-1} \qquad (41)$$

**Derivation of the Mean Squared Error** The Mean Squared Error (MSE) is the total squared error divided by the total number of predicted elements, $m_b d$. The prediction error matrix is $E = Y_b - \hat{W} Y_a$.

$$E \approx A_b Z - \left( A_b A_a^\mathsf{T} (A_a A_a^\mathsf{T})^{-1} \right) A_a Z \qquad (42)$$

$$= \left( A_b - A_b A_a^\mathsf{T} (A_a A_a^\mathsf{T})^{-1} A_a \right) Z \qquad (43)$$

The total squared error is the squared Frobenius norm of $E$.

$$\|E\|_F^2 = \text{Tr}(E^\mathsf{T} E) \approx \text{Tr}\left( Z^\mathsf{T} (\ldots)^\mathsf{T} (\ldots) Z \right) \qquad (44)$$

$$= \text{Tr}\left( (\ldots)^\mathsf{T} (\ldots) (Z Z^\mathsf{T}) \right) \quad \text{(using cyclic property of trace)}$$

$$\approx d \cdot \text{Tr}\left( (\ldots)^\mathsf{T} (\ldots) \right) = d \left\| A_b - A_b A_a^\mathsf{T} (A_a A_a^\mathsf{T})^{-1} A_a \right\|_F^2$$

Dividing the total squared error by $m_b d$ yields the final MSE expression:

$$\text{MSE}(Y_b | Y_a) \approx \frac{1}{m_b} \left\| A_b - A_b \left( A_a^\mathsf{T} (A_a A_a^\mathsf{T})^{-1} A_a \right) \right\|_F^2 \qquad (45)$$

Notation:

$$\hat{Y}_b = (\hat{y}_{b,(1)}, \ldots, \hat{y}_{b,(d)}) \qquad (46)$$

$$\mathrm{E}[\hat{Y_b}^i] \equiv \frac{1}{d} \sum_{k=1}^{d} \hat{y}_{b,(k)}^i = \frac{1}{d} \sum_{k=1}^{d} \sum_m \hat{W}^{im} y_{a,(k)}^m$$

$$= \frac{1}{d} \sum_{k=1}^{d} \sum_{m,n} \hat{W}^{im} A_a^{mn} z_{(k)}^n = \sum_{m,n} \hat{W}^{im} A_a^{mn} \frac{1}{d} \sum_{k=1}^{d} z_{(k)}^n$$

$$\approx \sum_{m,n} \hat{W}^{im} A_a^{mn} \mathrm{E}[z^n]$$

$$= 0$$

$$\mathrm{E}[y^i y^j] = \sum_{m,n} A^{im} A^{jn} \mathrm{E}[z^m z^n] = \sum_{m,n} A^{im} A^{jn} \delta_{mn} = \sum_m A^{im} A^{jm} = (AA^\mathsf{T})_{ij}$$

**Derivation of the Explained Variance $R^2$** The Explained Variance $R^2$ is defined by:

$$R^2 = 1 - \frac{SS_\mathrm{res}}{SS_\mathrm{tot}} \tag{47}$$

where

$$SS_\mathrm{res} = \sum_{k=1}^{d} ||y_{b,(k)} - \hat{y}_{b,(k)}||^2 \tag{48}$$

$$SS_\mathrm{tot} = \sum_{k=1}^{d} ||y_{b,(k)} - \bar{y}_b||^2 \tag{49}$$

$$\bar{y}_b = \frac{1}{d} \sum_{k=1}^{d} y_{b,(k)} \tag{50}$$

We can derive an analytical expression of $SS_\mathrm{res}$, $SS_\mathrm{tot}$, and $\bar{y}_b$ in terms of the projection matrices $A_a$ and $A_b$:

$$\bar{y}_b = \frac{1}{d} \sum_{k=1}^{d} y_{b,(k)} = A_b \frac{1}{d} \sum_{k=1}^{d} z_k \approx A_b \mathrm{E}[z] \tag{51}$$

$$= 0 \tag{52}$$

$$SS_\mathrm{res} = \sum_{k=1}^{d} ||y_{b,(k)} - \hat{y}_{b,(k)}||^2 = \mathrm{Tr}[(Y_b - \hat{Y}_b)^\mathsf{T}(Y_b - \hat{Y}_b)] = \mathrm{Tr}[Z^\mathsf{T}(A_b - \hat{W}A_a)^\mathsf{T}(A_b - \hat{W}A_a)Z] \tag{53}$$

$$= \mathrm{Tr}[(A_b - \hat{W}A_a)^\mathsf{T}(A_b - \hat{W}A_a)ZZ^\mathsf{T}] \approx d \cdot \mathrm{Tr}[(A_b - \hat{W}A_a)^\mathsf{T}(A_b - \hat{W}A_a)] \tag{54}$$

$$SS_\mathrm{tot} = \sum_{k=1}^{d} ||y_{b,(k)} - \bar{y}_b||^2 \approx \sum_{k=1}^{d} ||y_{b,(k)}||^2 = \mathrm{Tr}[Y_b^\mathsf{T} Y_b] \tag{55}$$

$$= \mathrm{Tr}[Z^\mathsf{T} A_b^\mathsf{T} A_b Z] = \mathrm{Tr}[A_b^\mathsf{T} A_b ZZ^T] \tag{56}$$

$$\approx d \cdot \mathrm{Tr}[A_b^\mathsf{T} A_b] \tag{57}$$

Thus the analytical expression of $R^2$ can be expressed as:

$$R^2 = 1 - \frac{SS_{\text{res}}}{SS_{\text{tot}}} = 1 - \frac{\text{Tr}[(A_b - \hat{W}A_a)^{\mathsf{T}}(A_b - \hat{W}A_a)]}{\text{Tr}[A_b^{\mathsf{T}}A_b]} \tag{58}$$

**Derivation of the Pearson Correlation**    The prediction is $\hat{Y}_b = \hat{W}Y_a$

The Pearson Correlation matrix between the prediction and the ground truth is given by:

$$\rho(\hat{Y}_b, Y_b)_{ij} \equiv \rho(\hat{Y}_b^{\,i}, Y_b^{\,j}) = \frac{\text{Cov}(\hat{Y}_b^{\,i}, Y_b^{\,j})}{\sqrt{\text{Var}(\hat{Y}_b^{\,i})\text{Var}(Y_b^{\,j})}} \tag{59}$$

Where indices $i$ and $j$ correspond to system dimensions. The Covariances can be expressed as:

$$\text{Cov}(\hat{Y}_b^{\,i}, Y_b^{\,j}) = \frac{1}{d-1}\sum_{k=1}^{d}\hat{y}_{b,(k)}^i y_{b,(k)}^j = \frac{1}{d-1}\sum_{k=1}^{d}\sum_{m}\hat{W}^{im}y_{a,(k)}^m y_{b,(k)}^j$$

$$= \frac{1}{d-1}\sum_{k=1}^{d}\sum_{m,n,l}\hat{W}^{im}A_a^{mn}z_{(k)}^n A_b^{jl}z_{(k)}^l = \frac{1}{d-1}\sum_{m,n,l}\hat{W}^{im}A_a^{mn}A_b^{jl}\sum_{k=1}^{d}z_{(k)}^n z_{(k)}^l$$

$$\approx \frac{1}{d-1}\sum_{m,n,l}\hat{W}^{im}A_a^{mn}A_b^{jl}d\cdot\text{E}[z^n z^l] = \frac{d}{d-1}\sum_{m,n,l}\hat{W}^{im}A_a^{mn}A_b^{jl}\delta_{nl}$$

$$\approx \sum_{m,n}\hat{W}^{im}A_a^{mn}A_b^{jn} = (\hat{W}A_a A_b^{\mathsf{T}})_{ij} = (A_b A_a^{\mathsf{T}}(A_a A_a^{\mathsf{T}})^{-1}A_a A_b^{\mathsf{T}})_{ij}$$

And the Variances:

$$\text{Var}(\hat{Y}_b^{\,i}) = \frac{1}{d-1}\sum_{k=1}^{d}\hat{y}_{b,(k)}^i \hat{y}_{b,(k)}^i = \frac{1}{d-1}\sum_{k=1}^{d}\sum_{m,n}\hat{W}^{im}y_{a,(k)}^m \hat{W}^{in}y_{a,(k)}^n$$

$$= \frac{1}{d-1}\sum_{m,n}\hat{W}^{im}\hat{W}^{in}\sum_{k=1}^{d}y_{a,(k)}^m y_{a,(k)}^n$$

$$\approx \frac{1}{d-1}\sum_{m,n}\hat{W}^{im}\hat{W}^{in}d\cdot\text{E}[y_a^m y_a^n]$$

$$= \frac{d}{d-1}\sum_{m,n}\hat{W}^{im}\hat{W}^{in}(A_a A_a^T)_{mn}$$

$$\approx (\hat{W}(A_a A_a^{\mathsf{T}})\hat{W}^{\mathsf{T}})_{ii}$$

$$= (A_b A_a^{\mathsf{T}}(A_a A_a^{\mathsf{T}})^{-1}(A_a A_a^{\mathsf{T}})\hat{W}^{\mathsf{T}})_{ii}$$

$$= (A_b A_a^{\mathsf{T}}(A_a A_a^{\mathsf{T}})^{-1}A_a A_b^{\mathsf{T}})_{ii}$$

$$\text{Var}(Y_b^{\,j}) = \frac{1}{d-1}\sum_{k=1}^{d}y_{b,(k)}^j y_{b,(k)}^j \approx \frac{d}{d-1}\text{E}[y_b^j y_b^j]$$

$$\approx (A_b A_b^{\mathsf{T}})_{jj}$$

Expressed in $A_a$ and $A_b$, the Pearson Correlation matrix becomes:

$$\rho(\hat{Y}_b, Y_b)_{ij} \approx \frac{(A_b A_a^{\mathsf{T}}(A_a A_a^{\mathsf{T}})^{-1}A_a A_b^{\mathsf{T}})_{ij}}{\sqrt{(A_b A_a^{\mathsf{T}}(A_a A_a^{\mathsf{T}})^{-1}A_a A_b^{\mathsf{T}})_{ii}(A_b A_b^{\mathsf{T}})_{jj}}} = \frac{(\hat{W}A_a A_b^{\mathsf{T}})_{ij}}{\sqrt{(\hat{W}A_a A_b^{\mathsf{T}})_{ii}(A_b A_b^{\mathsf{T}})_{jj}}} \tag{60}$$

