# OpenReview forum: "Mirages of Misalignment: How Superposition Distorts Neural Representation Geometry"
_ICLR.cc/2026/Conference — Submitted to ICLR 2026_

### Official Review · Reviewer_DJPr · 2025-10-24

**Soundness:** 2
**Presentation:** 2
**Contribution:** 1
**Rating:** 2
**Confidence:** 4

**Summary:**

This work argues that although representations of models might be dissimilar when using linear measures of similarity
like Representational Similarity Analysis (RSA), the models might still have captured the same latent features.
The paper says that it might be possible to find these features using sparse autoencoders (SAEs).

They do an experiment on latent recovery.

They also do an experiment on whether using SAEs to make higher dimensional representations can give higher RSA and
linear regression prediction between penultimate layers of models, between layers 1-4 of a ResNet-50 model and a brain
and between two brains.

They find that RSA is higher between the SAE representations than between the original model representations,
while linear regression is about the same.
RSA also increases for the model to brain comparison. Linear regression increases in some cases.
For brain to brain there was no change.

**Strengths:**

**S1:** Work on representational similarity is very relevant for both representation learning and neuroscience.

**S2:** It is true that it is possible to have two models with the same m latent variables, but where their lower dimensional
(dimension of n < m) representations are not related in a linear manner.

**Weaknesses:**

**W1:** There is a flaw in the motivation for this paper. Using measures which measure linear similarity between model
representations are not "systematically misleading" (line 22) and do not
"erroneously measure their representations as dissimilar" (line 53). They simply measure linear similarity of the representations.
See question **Q1**.


**W2:** The evaluation in the experiments is not thorough enough to make any conclusions. It seems only one seed of every model is compared.
It also seems only one seed of SAE is trained for each hyperparameter choice and only the best one is reported. See questions **Q4,Q5**.



**W3:** The paper assumes that the neural representations are a linear map of the latent variables. This assumption is too strong,
since neural networks allow for non-linear functions. At the very least a good argument should be given for why this is reasonable.
See question **Q2**.


**W4:**  There are some unsupported claims:
- Line 16-18 in the abstract: "We derive an analytic theory that predicts this apparent misalignment for common linear metrics like
representational similarity analysis and linear regression".
However, the theory does not predict misalignment, it only says that misalignment is possible.

- In line 41-43 it says "even when models are trained on identical tasks and data, comparisons consistently
reveal a persistent “alignment ceiling,”" and this is followed by three references [1], [2], [3]. However, none of these references
talk about an "alignment ceiling" for models trained on the same data. [1] is about how similar model representations are with the brain,
[2] considers alignment of alignment metrics, and [3] is again alignment between models and the brain. This claim is repeated with the
same references in line 112-114.


References:

[1] Martin Schrimpf, Jonas Kubilius, Ha Hong, Najib J Majaj, Rishi Rajalingham, Elias B Issa, Kohitij Kar, Pouya Bashivan,
Jonathan Prescott-Roy, Franziska Geiger, et al.
Brain-score: Which artificial neural network for object recognition is most brain-like? BioRxiv, pp. 407007, 2018.


[2] Jannis Ahlert, Thomas Klein, Felix Wichmann, and Robert Geirhos. How aligned are different alignment metrics?
arXiv preprint arXiv:2407.07530, 2024.


[3] Zirui Chen and Michael F Bonner. Universal dimensions of visual representation. Science Advances,
11(27):eadw7697, 2025.

**Questions:**

Since the motivation is flawed (**W1**) and both the theoretical(**W3,W4**) and experimental(**W2**) contributions are lacking, I recommend rejection.



**Questions:**

**Q1:** Consider two vectors: (100, 0) and (1, 0.1).
If we use Euclidean distance to measure their difference, we get a high value, so they are dissimilar with respect to Euclidean distance.
If we now normalize the vectors to have length 1 and then measure their difference with Euclidean distance, we get a quite small value.
So the normalized vectors are similar with respect to Euclidean distance. This is not because the measure was "wrong" in the first case
and the vectors are actually similar, it is because we are measuring two different things.
In the same way, if we use representational similarity analysis (RSA) to measure the similarity between representations of two models
and get a low value, then the representations are dissimilar with respect to RSA. If we train sparse autoencoders (SAEs) to make higher
dimensional representations and get a higher similarity when we compare these new representations, then that does not mean that RSA
was wrong about the original representations being dissimilar. It only means that we are now measuring a different thing.

If you want to argue that similarity between the representations from the SAEs is the similarity we should care
about when comparing models, then that is fine, but it does require an argument. So why is it the similarity between the representations
from the SAEs which is important when comparing models?


**Q2:** Definition 3.1 and assumption 1 (line 151-152):
The paper assumes that the neural representations are a linear map of the latent variables. However, neural networks
allow for non-linear functions. Why is this a reasonable assumption to make?


**Q3:** Line 156: What is your definition of $\Vert \cdot \Vert_0 $?


**Q4:** Line 354-356: Did you only train one seed for each choice of hyperparameters?


**Q5:** Line 358-359: "Alignment is taken over each SAE architecture and we report the SAE with the highest mean alignment increase
for each metric." Does this mean that you trained multiple SAEs and only report the best one?


**Q6:** Figure 4: With respect to the linear regression plot: Does this mean that you get better linear regression results when you
predict the brain representations from the model compared to when you predict the model representations from the brain?







**Additional Feedback:**

typos:


Line 32: question mark in reference.


Line 50-51: "to extract the representations" -> "to extract the features"


Line 84: question mark in reference.


Line 95-96: "Linear Regression performed with these recovered features" -> "Linear Regression performed better with these recovered features"

---

> ### Author Response · Authors · 2025-11-25
> **Interim update**
>
> Thank you for your helpful suggestions. We agree that feature-based alignment captures different information than standard alignment metrics and appreciate the opportunity to clarify and better motivate our study. We are open to additional suggestions if our ongoing experiments do not address your concerns thus far. Relevant sections of the current paper are highlighted in red.
>
> 1. **Noted unique brain result:** *“For brain to brain there was no change.”* **Response:** There was indeed no positive change in feature space for brain-brain comparisons in the initial draft. This persists, and through independent feature verification and improved model selection criteria, we choose a model that actually exacerbates the decrease in alignment for brains (especially higher level areas). In the current draft we address this intriguing phenomenon directly. Section 5 / Figure 5 shows that inductive biases more prevalent in the brain in general, and higher areas of brain/model, can drive up RSA in the neural space, reducing it in feature space. Feature space alignment allows us to see the change in features associated with reduced alignment.
> 2. **Provide appropriate context for feature-based alignment:** *”Using measures which measure linear similarity between model representations are not "systematically misleading" (line 22) and do not "erroneously measure their representations as dissimilar" (line 53). They simply measure linear similarity of the representations. See question Q1. + “why is it the similarity between the representations from the SAEs which is important when comparing models?”* **Response:** We agree and have reframed the language in lines 22 and 53, as well as the introduction in general. In the current framing we clarify that the similarity between the representations from the SAEs (and even comparing the neural to SAE space transition in similarities) can help alleviate the dimensionality bias observed using standard similarity metrics, and at least help directly observe the factors driving increased similarity in the neural space. We show that changes in alignment from neurons -> latents coincide with changes in the features represented and encoded by the neural networks (Figures 6,9,10).
> 3. **Strengthen SAE training and selection process:** *“It seems only one seed of every model is compared. It also seems only one seed of SAE is trained for each hyperparameter choice and only the best one is reported. See questions Q4,Q5. +  Did you only train one seed for each choice of hyperparameters? + Does this mean that you trained multiple SAEs and only report the best one?”* **Response:** We now explicitly choose the most disentangled SAE model out of 100 trained models using the Unsupervised Disentanglement Ranking metric, which selects the SAE with the lowest pairwise distance to all other trained models as in Higgins et al. (2021). We validate that SAE latents recover disentangled features using an automated interpretability metric, which evaluates the semantic consistency of each features’ preferred images. More details can be found in the new section 4.5: Model Selection and Validation.
> 4. **Address linear representation assumption:** *“ The paper assumes that the neural representations are a linear map of the latent variables. This assumption is too strong, since neural networks allow for non-linear functions. At the very least a good argument should be given for why this is reasonable. See question Q2.”* **Response:** We appreciate this concern and address the linearity assumption as a limitation of scope in our Limitations section. While we acknowledge that neural network analysis may benefit from nonlinear metrics, we state that many features are in fact linearly represented and recoverable. Recent successes in linear probing and SAE-based methods in large models, combined with experiments supporting the superposition hypothesis (Elhage et al., 2022), demonstrate that linear feature representations are pervasive enough to warrant investigation.
> 5. **Confirm source- target result:** *"With respect to the linear regression plot: Does this mean that you get better linear regression results when you predict the brain representations from the model compared to when you predict the model representations from the brain?"* **Yes**

---

> > ### Comment · Reviewer_DJPr · 2025-11-25
> >
> > Thank you for your answers and the new version of the paper.
> >
> > I would like to note that I find the updated paper very different from the original,
> > since all figures except figure 1 have been changed, and I would guess at least 40% of the text has been changed,
> > reflecting e.g. the new experiments. All in all, I would say that these changes are more than what one should make in a discussion phase.
> >
> > With respect to the weaknesses in the new paper, there are the following changes:
> >
> >
> > **W1:** The new paper does not seem to have this weakness.
> >
> > **W2:** Picking the SAEs to compare based on a disentanglement metric instead of just the ones showing the highest alignment
> > would be a step in the right direction. However, in line 349-351 it says: "The model with the highest average pairwise similarity
> > to all others receives the highest UDR score and is considered most disentangled, as it represents the most consistent solution
> > across the optimization landscape." This is not what "disentangled" usually means. Usually it means that the individual features
> > or latents are independent of each other.
> >
> > Also, the experiments in figure 2 and 4 show no increase in similarity using the latents instead of the neurons.
> > The experiments in figure 3 show increase for some, but decrease for others.
> > This means the experiments do not support the claim made in the abstract (line 21-22 "we find that feature-based alignment reveals higher similarity")
> > and the discussion (line 523-526 "we used SAEs and K-Means to extract approximations of features in real neural networks, showing that
> > alignment over latent activations is often significantly higher for the commonly used metrics of RSA and Linear Regression").
> > The claim in the introduction is more nuanced (line 103-105 "Applying feature-based alignment to real neural networks,
> > we find that disentanglement often increases alignment between systems, but also observe cases where relative alignment between
> > networks changes—with some networks becoming less similar in the latent space."), but I would still say that
> > "disentanglement often increases alignment between systems" is too strong a claim given the results presented.
> >
> > **W3:** The paper still assumes that the neural representations are a linear map of the latent variables.
> > I see this has been added to the limitations section (why is it not marked in red?).
> >
> > **W4:** The claim about the "alignment ceiling" has been removed.
> > The claim about predicting misalignment has been moved to line 17-18
> > "We formalize this through an analytic theory predicting apparent misalignment for common linear metric".
> > As mentioned before, the theory does not predict misalignment, it only says that misalignment is possible.
> >
> > **W5 (NEW):** There is a mismatch between the new experiments and the presented theory of superposition.
> > The paper now uses K-means to make latents for the brain to brain experiments.
> > How this should relate to superposition disentanglement is unclear.
> >
> > My suggestion to the authors is to rewrite the paper in a way such that the claims match the results, and such that the
> > experiments match the presented theory, and then send the resulting paper to another conference.

---

### Official Review · Reviewer_axwD · 2025-10-25

**Soundness:** 2
**Presentation:** 3
**Contribution:** 2
**Rating:** 2
**Confidence:** 4

**Summary:**

The paper examines the issue of representational similarity. The authors assume that neural representations are linear functions of some latent variables that have a higher dimensionality than the representation itself, and that are also sparse. Under this assumption, the authors prove that even if this latent representation is the same, two different linear functions of the same latent variable might result in very different representations according to the usual similarity measures (RSA and linear regression) if the linear mappings are different. The authors then present empirical results, which demonstrate that in some cases, the similarity of latent representations (extracted using a sparse autoencoder) is higher than that of the original raw representations.

**Strengths:**

The paper adds a new angle to the analysis of representation similarity, namely the consideration of latent variables. This has the potential to lead to new directions and insights. Typically, latent variables are currently not considered when comparing representations, so this aspect carries a certain novelty.

**Weaknesses:**

The key intuition, namely that models trained on the same data must have the same latent variables, and any differences result from different superpositions of these variables, is **extremely difficult to verify empirically**. To a certain extent, it is also a tricky assumption, because in a sense it is trivial: the representations in different models are computed from the same input, therefore they are indeed only different functions of the same "latent variables" (the input). Of course, the key is then to find latent representations that have a much lower dimensionality than the input (an aspect not considered in the paper), from which the representation is computed using a very simple function (eg, linear).

But even so, if one tries to support the intuitive assumption via finding such latent variables with eg SAEs, and the resulting latent representations are more similar, we still don't know whether this higher similarity is theoretically inevitable due to using the same inputs (in which case it is not interesting), or indeed we found something fundamentally interesting: a common latent representation distorted by superpositions. The paper does not attempt to address this question.

However, in the brain-brain experiment, RSA did not increase, maybe because the SAE was trained on different inputs? This seems to support the first explanation.

The theoretical analysis gives very little support as well, because even if the latent variables are the same, the models learn a linear mapping that is optimal for the task at hand, and **that is not likely to be a random mapping**. And we do not know whether functionally good mappings are similar or not, the paper does not address this.

Typos: missing ref in line 86. Paragraph title should be "brain to brain" and not "brain to model" in line 385.

**Questions:**

Please elaborate on why your methodology is suitable for testing whether your fundamental hypothesis that common latent representations are distorted by superposition? (And the higher similarity of latent representations is not simply an artifact of using higher dimensions and using the same dataset).

Let's assume that we are able to prove that you are right and latent representations tend to be more similar on the same task than neural representations. What is the practical implication? Isn't this an epiphenomenal statement?

---

> ### Author Response · Authors · 2025-11-25
> **Interim update**
>
> Thank you for your detailed comments. We agree that addressing them will strengthen the paper by providing more appropriate controls and highlighting the value of feature-based representational alignment, distinct from artifacts of dimensionality gains. We welcome feedback if the revisions thus far do not address your concerns. Relevant sections of the current paper are highlighted in red.
>
> 1. **Address inevitability of alignment bump:** *“we still don't know whether this higher similarity is theoretically inevitable due to using the same inputs (in which case it is not interesting), or indeed we found something fundamentally interesting: a common latent representation distorted by superpositions.”* **Response:** We agree that we did not highlight the intriguing cases of lower alignment in the latent space. Now, we point to examples in the real data where lower similarity is observed in the latent space (Ex: Figure 3 - Model-Brain comparisons, described in Section 4.6- Results, Model-Brain section). We hypothesize that this is due to shared inductive bias driving real similarity in the neural space. We simulate the effects of bias, among other things, in the new section 5 (Figure 5). We find that indeed there is more semantic categorical overlap between images preferred by neurons than latents for those networks that are more similar in the neural than latent space (Figure 6).
> 2. **Address causes of brain-brain result:**  *“However, in the brain-brain experiment, RSA did not increase, maybe because the SAE was trained on different inputs? This seems to support the first explanation.”* **Response:** Consistent with our above response, we interpret this finding (at least in higher level brain areas) is due to the alleviation of inductive bias driving representational similarity in the neural space. We also use K-Means in place of SAE, since this yielded better features for biological data as verified by automated interpretability measurements described in the new section 4.5: Model Selection and Validation. This is possibly because there is less ‘superposition’ in the brain or due to the resolution of the brain fMRI recordings. Finally, the current version leverages within-subject comparison so that we are able to train models over 9,000 more images.
> 3. **Explain random mappings assumption:** *“because even if the latent variables are the same, the models learn a linear mapping that is optimal for the task at hand, and that is not likely to be a random mapping.”* **Response:** Agreed. We simulate the effects of less random mappings (Section 5, Figure 5). Mainly, we address this assumption as a limitation in the discussion, and clarify that we make the assumption for practical purposes to test the sufficiency of superposition arrangements to induce geometric bias.
> 4. **Address confounds in alignment bump:** *“ elaborate on why your methodology is suitable for testing whether your fundamental hypothesis that common latent representations are distorted by superposition? (And the higher similarity of latent representations is not simply an artifact of using higher dimensions and using the same dataset).”* **Response:** We acknowledge that this methodology cannot be viewed as suitable without 1) independent verification that SAEs are learning meaningful features and 2) explicitly addressing dimensionality artifacts. We independently verify that SAEs learn meaningful features with automated interpretability, and by demonstrating that the changes in alignment observed in Figures 3-5+ coincide with changes in the preferred stimuli of the latents relative to the neurons (Figure 9, Figure 10). In the setting where all latent dimensions are fixed, we also address the lack of inevitability of increased alignment, as mentioned in an above response.
> 5. **Explain significance:**  *“What is the practical implication? Isn't this an epiphenomenal statement?”* **Response:** We now highlight (particularly in the modified introduction) that the practical implications are 1) more appropriate model selection, as the relative similarities of neural networks do shift in Figures 2-4 and 2) gaining a deeper (potentially actionable) understanding  of the true factors driving model-model, brain-model, and brain-brain divergence.

---

### Official Review · Reviewer_he2Q · 2025-10-30

**Soundness:** 3
**Presentation:** 3
**Contribution:** 2
**Rating:** 2
**Confidence:** 3

**Summary:**

The paper investigates why alignment metrics such as RSA and linear regression often yield low similarity even when networks encode identical features.
They proposed that this could arise from superposition — the tendency of neural networks to linearly represent more features than the number of neurons available.

They derive an analytic theory predicting how this feature superposition lowers alignment scores even when networks encode identical features. They then validate the prediction using (i) synthetic linear simulations and (ii) real-network experiments (ResNet, ViT, fMRI) with and without Sparse Autoencoder (SAE) preprocessing, showing that alignment increases after “de-superposing” representations via SAEs.

**Strengths:**

- The motivation is clear: the authors are trying to answer why the features learned by neural networks misalign even when networks encode identical features.
- They conduct experiments both on synthetic data and real-network experiments to show that alignment increases after using SAE to de-superpose.

**Weaknesses:**

- Some references couldn't be compiled: line 32, line 86
- The theoretical result is elementary: they are straightforward applications of random-projection algebra. No nontrivial mathematical results or bounds are proven.
- The SAE experiments are presented as confirmation of the theory, but no independent check is provided that SAE latents truly “recover” disentangled features.
The observed alignment gain could arise from dimensionality changes, normalization, or denoising rather than from resolving superposition.

**Questions:**

The paper focuses on alignment measured by linear metrics such as RSA and linear regression.
However, the features learned by neural networks are not necessarily linear, and representational similarity may depend on non-linear relationships.
- Have the authors considered evaluating alignment using other metrics, for example, kernel alignment, non-linear CCA, or mutual information measures
(see Huh et al., 2024; Insulla et al., 2025, Kornblith 2019,  Williams 2021, e.t.c)

The references i mentioned are:
- Huh et al 2024. The platonic representation hypothesis
- Insulla et al 2025. Towards a Learning Theory of Representation Alignment
- Kornblith et al., 2019. Similarity of Neural Network Representations Revisited. ICML
- Williams et al 2021. Generalized shape metrics on neural representations.

---

> ### Author Response · Authors · 2025-11-25
> **Interim update**
>
> Thank you for your helpful comments. In addition to independently verifying SAE features, we have incorporated your suggested citations and have more clearly defined our scope. We welcome additional feedback if our responses do not properly address your points. Relevant sections of the current paper are highlighted in red.
>
> 1. **Verify SAEs learn meaningful features:** *“no independent check is provided that SAE latents truly “recover” disentangled features."* **Response:** We now explicitly choose the most disentangled SAE model out of 100 trained models using the Unsupervised Disentanglement Ranking (UDR) metric, which selects the SAE with the lowest pairwise distance to all other trained models as in Higgins et al. (2021). We validate that SAE latents recover disentangled features using an automated interpretability metric, the odd-one-out task, which evaluates the semantic consistency of each feature's or neuron's preferred images. More details can be found in the new Section 4.5: Model Selection and Validation.
> 2. **Address nonlinear metrics:** *“Have the authors considered evaluating alignment using other metrics, for example, kernel alignment, non-linear CCA, or mutual information measures (see Huh et al., 2024; Insulla et al., 2025, Kornblith 2019, Williams 2021, e.t.c)"* **Response:** We appreciate this suggestion and agree that nonlinear metrics offer valuable complementary perspectives. Our focus on linear metrics is motivated by recent evidence that many features are linearly represented and recoverable (superposition hypothesis; Elhage et al., 2022). We acknowledge this scope limitation in our revised manuscript and explicitly note that the success of nonlinear metrics suggests they may uncover additional sources of alignment beyond what linear methods reveal. If helpful for the revision, we would be happy to incorporate mean-centered Euclidean RSA following Williams et al. (2024), which provides results with higher correspondence to nonlinear CKA.

---

### Official Review · Reviewer_xRMD · 2025-10-31

**Soundness:** 2
**Presentation:** 3
**Contribution:** 3
**Rating:** 4
**Confidence:** 3

**Summary:**

The paper studies whether feature superposition lowers standard linear alignment measures between representations. It gives simple analyzable models that predict drops in RSA and linear-regression alignment due to superposition, and then shows that “demixing” with sparse autoencoders tends to increase those alignment scores in artificial neural networks. Experiments span simulations, model to model, model to brain, and brain to brain.

**Strengths:**

- Clear and novel central hypothesis that raises an interesting point. I would say their work is original.

- Theory is interpretable and connects directly to quantities people already report in alignment papers.

- SAE experiments are interesting and well-designed (although I believe they can be improved by providing more experimental results)

**Weaknesses:**

I believe the significance can be improved by providing more experimental results (more model-model comparisons, baseline comparisons). I am a bit cautious about whether to accept the SAE experimental result as strong evidence of the author's claim that superposition is the reason why we see lower than expected RSA/LR measurements. This is mainly because the authors did not check whether their SAEs actually learned meaningful features and whether the same features are found across different models.

I think addressing the following points would strengthen the significance:

- Without evaluating the features learned in SAE, the SAE cannot be claimed to have disentangled superposition. As of now, without the analysis, the author's SAE is nothing more than just some nonlinear expansion to me. An increase in alignment after a nonlinear expansion, in general, is not very surprising, unless this nonlinear expansion is really due to the disentanglement of superimposed features. I think it will be most convincing if the authors can find common SAE features across models trained on a single dataset.

- Measuring alignments in different kinds of expansion and comparing them to the SAE expansion. If SAE expansion has a greater increase in alignment than other nonlinear expansions do, then that would strengthen the authors' claim. As a simple example, the similarity can be measured at (1) the initialization of the SAE, (2) kernel space (LR can be simply replaced with kernel ridge regression; for example, the authors can just use kernel ridge regression with RBF kernel). I am not sure if there is a kernel version of RSA, but even if it does not exist, the author can use a random feature model (like the initialization of SAE). As a note, the random feature version of RBF kernel is the random Fourier feature. These other nonlinear expansions can serve as baselines.


I should highlight that I find the hypothesis of this paper and its preliminary findings very intriguing and thought-provoking. I am willing to increase my score if my concerns can be addressed.

**Questions:**

1. The asymptotic RSA alignment formula (Theorem 4.1) takes the form of CKA, in terms of the maps A_a and A_b (so not in terms of features like the typical CKA). Interestingly, I think the same formula will appear in the asymptotic limit of CKA: i.e. CKA(Y_a,Y_b) -> CKA(A_a,A_b) (which is the RHS eqn. 5). If that is the case, then the authors can also claim the CKA is also affected by superposition (which is not very surprising, but still it would be a nice additional side claim). Can the authors confirm this?

2. I think the authors should clearly state what they mean by "asymptotic" limit in the main text as well, when they explain the theory.

3. There are missing citations in the main text (shown as "?" in the pdf)

4. Line 103 "A central goal in neuroscience and machine learning is to compare learned representations across different system" is an odd claim to make.

5. The notations M and m are used interchangeably. Please stick with one for consistency. (the same applies to N and n)

6. Compression is not defined in the main text. It is only defined as N/M in the figure caption, but that definition confuses me. In Figure 2, a compression factor <1 is highlighted as "No CS" but this implies that M=N is the CS threshold, which is wrong. Instead, M=K log(N/K) should be the CS threshold as stated in the main text. Shouldn't the compression factor  be defined as K log(N/K) /M for the CS/No CS highlighted areas to make sense in Figure 2?

7. Line 385 Brain to Brain, not brain to model

8. Why in Figure 2 do the authors report R^2 for LR, but in Figure 3, the authors report Pearson R?

9. In Figure 5, I think "Increase in RSA over features" should be corrected to "Decrease in RSA over features"

---

> ### Author Response · Authors · 2025-11-25
> **Interim update**
>
> Thank you for your thoughtful suggestions. We provide the following update on the steps we have taken to address them so far and would welcome your thoughts on whether these revisions adequately address your listed concerns. Relevant sections of the current paper are highlighted in red.
>
> 1. **Verify SAEs learn meaningful features:**  *“the authors did not check whether their SAEs actually learned meaningful features and whether the same features are found across different models.”* **Response:** We now explicitly choose the most disentangled SAE model out of 100 trained models using the Unsupervised Disentanglement Ranking (UDR) metric, which selects the SAE with the lowest pairwise distance to all other trained models as in Higgins et al. (2021). We validate that SAE latents recover disentangled features using an automated interpretability metric, the odd-one-out task, which evaluates the semantic consistency of each feature's or neuron's preferred images. More details can be found in the new Section 4.5: Model Selection and Validation.
> 2. **Address alignment increase confound:** *“An increase in alignment after a nonlinear expansion, in general, is not very surprising,”* **Response:** Agreed. In addition to showing that our models learn meaningful features, we now highlight cases where we observe decreases in alignment after the nonlinear expansion when two systems share inductive biases (Section 5/Figure 5). For example, ResNet-50 Layer 4 and the face-selective region FFA-1 become less similar in the latent space. We show that cases of decreased latent alignment coincide with decreases in semantic categorical overlap of the preferred features of latents relative to neurons (Figure 6). Additionally, we address cases where the relative alignment between networks shifts and elaborate on the implications for model selection.

---

### Meta-Review · Area_Chair_NULr · 2025-12-10

**Summary:**

The paper proposes to apply feature-based alignment via SAEs/$k$-Means to reveal true similarity which is less affected by superposition.
The reviewers recognized the potential significance of this work --- it presents a clear, well-motivated hypothesis to explain misalignment of learned representations. The reviewers are concerned whether SAEs/$k$-means actually recover disentangled features, which requires more extensive empirical studies and broader alignment metrics. On the theory side, the reviewers emphasized the lack of in-depth theory to support the claims. Both concerns are valid. Overall, this work is interesting and could be a worthy contribution if the raised weaknesses are sufficiently addressed. The authors are encouraged to advance this work on both tracks.

**Reviewer Concerns:**

The authors enhanced the empirical pipeline (e.g. choosing the most consistent SAE among trials) during the rebuttal, which is appreciable. However, there is no guarantee the learned features are in the same representation space. Therefore the proposed method does not guarantee the recovery of shared features. The theoretical weakness is unfortunately not addressed, as it requires new developments through a major revision.

**Reviewer Scores:**

I could not identify evidence that reviewers' scores could raise significantly, especially noting the missing response to Reviewer DJPr and the overall brevity of the author's rebuttal.

---

### Decision · Program_Chairs · 2026-01-26

Reject